# Solar-driven direct air capture to produce sustainable aviation fuel

Yide Han[1], Olajide Otitoju [1], Ariane D. N. Kamkeng[1], Meihong Wang [1] ✉, Hui Yan[2], Fisher Millard[3], Wenli Du[4] ✉ & Feng Qian[4] ✉

Renewable energy-powered direct air capture with subsequent utilisation offers a sustainable decarbonisation strategy for a circular economy. Whereas current liquid-based capture technology relies on natural gas combustion for high-temperature calcination, restricting the transition to fully renewable operation. In this study, we show a 1MtCO$_2$/year solar-driven process that adopts a hydrogen fluidised solar calciner with onsite catalytic conversion of CO$_2$ into sustainable aviation fuel. We find that replacing fossil-fuel heating with solar thermal energy lowers electricity consumption by 63% and reduces onsite CO$_2$ emissions by 59%. The analysis shows that the production cost of sustainable aviation fuel is cost-effective (US\$4.62/kg) compared to the conventional process. Geographical sensitivity analysis indicates favourable deployment locations are low-risk countries with high solar irradiance and low hydrogen cost. The predicted results also outline potential economic viability for policymakers and industry investors.

Global warming has intensified the need for carbon dioxide removal (CDR) to achieve net-zero emissions by mid-century[1]. Direct air capture (DAC), which captures CO$_2$ from the atmosphere, is a key CDR approach due to its small land footprint and straightforward carbon accounting[2]. It is particularly effective for capturing legacy CO$_2$ in the air and balancing emissions from hard-to-abate sectors and heavy-duty long-haul transportation. According to the International Energy Agency (IEA), DAC is expected to capture over 85 million metric tonnes (Mt) of CO$_2$ by 2030, rising to 980 Mt by 2050, with one-third of this captured CO$_2$ projected to be utilised[3]. Compared to direct air carbon capture and storage (DACCS), which is cost-intensive and heavily dependent on policy incentives, direct air carbon capture and utilisation (DACCU) presents a potential for commercial use of CO$_2$. DACCU uses captured CO$_2$ to produce valuable chemicals or synthetic fuels such as sustainable aviation fuel (SAF)[4,5]. This approach provides a circular economy by recycling valuable materials rather than storing them in deep reservoirs[6–8].

CO$_2$ emissions from the aviation industry are responsible for 10% of transportation emissions and 2.5% of global emissions (1.03 Gt CO$_2$ in 2019)[9]. As air travel becomes more prevalent, aviation CO$_2$ emissions could reach roughly 2.0 Gt CO$_2$ by 2050, highlighting the urgent need for decarbonisation[9]. Replacing conventional aviation fuel with batteries or renewable energy is impractical in the short term because aircraft rely on energy-dense liquid fuel[10]. However, SAF is emerging as the most promising solution to meet aviation energy needs and reduce greenhouse gas (GHG) emissions due to its high energy density and drop-in nature[10]. In this respect, the application of CO$_2$ captured by DAC as low-carbon feedstock to produce drop-in SAF at large-scale[11,12] is regarded as an exciting option to fuel future aircrafts[13,14]. Besides, CO$_2$ use in the aviation sector could drive down costs and provide a market for DAC. For example, processes like AIR TO FUELS™ from Carbon Engineering (CE) Ltd and feed-to-liquid (XTL) by Shell are already being explored to generate feasible solutions[15,16].

Though still in its infancy, DACCU holds immense potential due to three growing areas: DAC, green hydrogen production, and sustainable fuel synthesis. The liquid-based DAC (L-DAC) process developed by CE (Fig. 1a) stands out for its relatively low energy consumption (5.25–8.81 GJ/t$_{CO2}$) and CO$_2$ capture cost (US\$94-712/t$_{CO2}$),

[1]Department of Chemical and Biological Engineering, The University of Sheffield, Sheffield, UK. [2]Department of Electrical and Electronic Engineering, University of Manchester, Manchester, UK. [3]Net Zero Energy, AtkinsRéalis, Edinburgh, UK. [4]Key Laboratory of Advanced Control and Optimization for Chemical Process of the Ministry of Education, East China University of Science and Technology, Shanghai, China. ✉e-mail: meihong.wang@sheffield.ac.uk; wldu@ecust.edu.cn; fqian@ecust.edu.cn

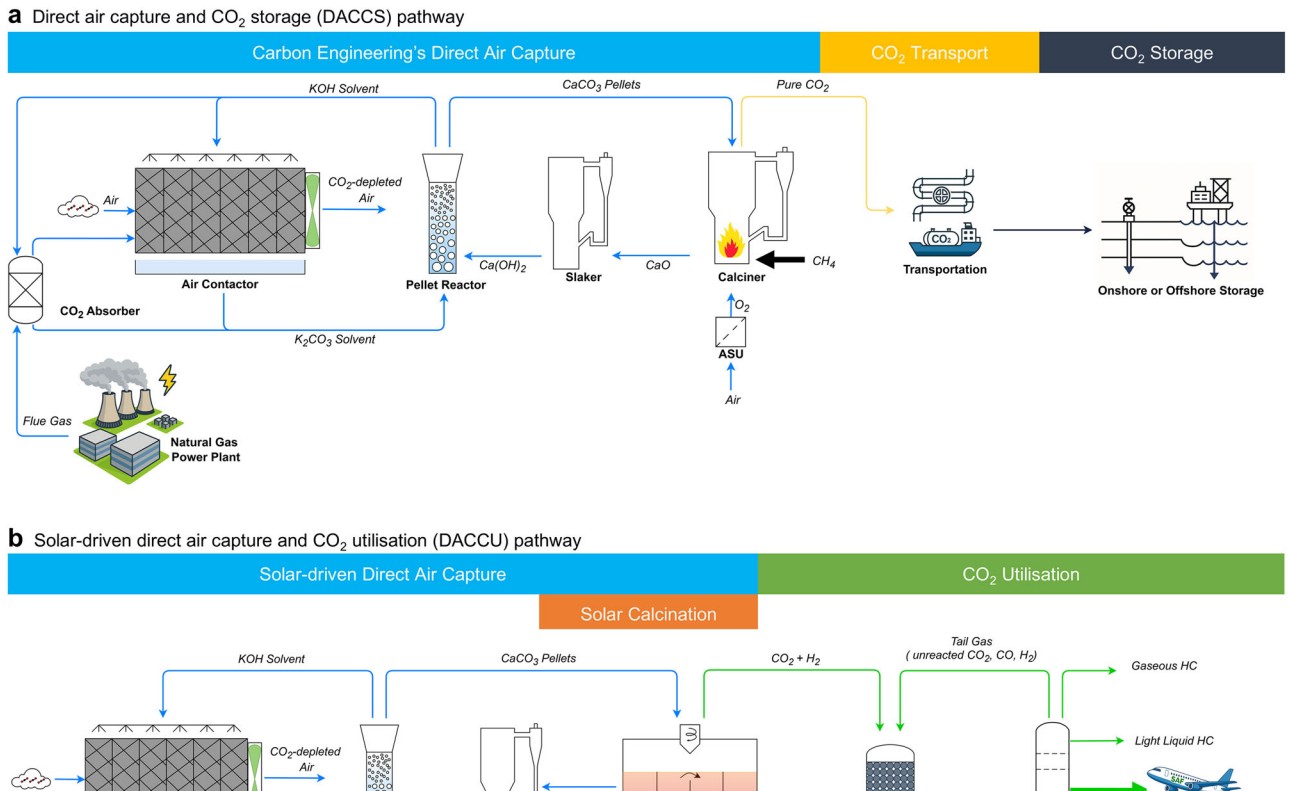

**Fig. 1 | Schematic representation of DAC for CO₂ storage or utilisation pathways. a** DAC (based on Carbon Engineering technology) and CO₂ storage (DACCS) pathway, where CO₂ is captured by DAC, transported via pipelines or ships, and stored underground or in the deep sea. **b** Solar-driven DAC and CO₂ utilisation (DACCU) pathway, incorporating solar-driven DAC and CO₂ utilisation sections. In the conventional DAC, electricity and heat demands are met by natural gas combustion, whereas in the proposed pathway, these demands are supplied by solar energy. Details of DACCU process design are presented in Supplementary Note 1. CSP concentrated solar power, HC hydrocarbon.

outperforming other amine-based and solid-based approaches[3,17–22]. It is adapted from existing commercial units and is currently at the demonstration stage (technology readiness level 7–8)[23], with Mt scale plants under construction in the USA and UK[24]. This technology faces challenges, primarily due to its reliance on natural gas combustion for electricity and thermal energy. The high-temperature calcination (800–900 °C), which accounts for over 90% of total energy consumption is a major contributor to life cycle CO₂ emissions[22] (Fig. 2a). To capture and store 1 t of atmospheric CO₂, 0.58 t (+0.2/−0.03 t) CO₂-equivalent emissions would be released, which partially offsets the captured CO₂[25]. Therefore, using renewable energy to supply high-temperature heat could maximise carbon removal potential and associated revenue streams[18,25,26].

CO₂-derived synthetic fuels can be produced via CO₂ hydrogenation[27]. Since the CO₂ is thermodynamically stable, hydrogenation of CO₂ usually favours the short-chain hydrocarbons (C₁–C₄). Traditionally, CO₂ is first converted to CO or methanol, which is then processed into liquid fuels. With the recent advances in catalysts[28,29], the direct route is formed by combining the reverse water-gas shift (RWGS) and Fischer-Tropsch synthesis (FTS) reactions. This direct pathway is well-suited for industry applications due to its ease of operation and cost-effectiveness compared to the traditional indirect route[5,30–32]. Laboratory-scale demonstrations of CO₂-to-jet fuel have achieved CO₂ conversion of 10–55% and a jet fuel yield of 6–18%[28,29,33,34].

Major obstacles lie in achieving high jet fuel selectivity due to the complex reaction mechanisms and the generation of large amounts of water[27]. Therefore, it is imperative to improve the process efficiency. This can be achieved by designing multifunctional catalysts with high selectivity and implementing advanced operating strategies[27,35].

As such, this study is based on L-DAC and one-step CO₂-FTS to develop a sustainable large-scale SAF production route. The DAC process is driven by renewable energy at affordable costs and integrated with CO₂ utilisation that yields a high SAF output. While several studies have explored solar-powered DAC, such as solar cells or alternative low-temperature CO₂ desorption methods[36,37], options for high-temperature processes remain limited[18]. Concentrated solar energy, also known as concentrated solar power (CSP), has shown the potential to provide high-temperature heat for solar calcination in L-DAC[38–41]. Scaling up these processes to megawatt levels presents challenges, particularly when using fixed-bed reactors—performance at large scales is restricted by heat transfer rates and temperature uniformity within the reactor bed[42,43]. However, applying a fluidised bed reactor can overcome these limitations, offering the potential for continuous operation and industrial scalability. As reported, the fluidised bed for solar calcination has been achieved at the world's largest 1-MW Odeillo's solar furnace, located in the French PROMES laboratory[44,45]. In this setup, a pilot-scale solar fluidised bed reactor is fluidised by air, with solar energy being transferred from sunlight to

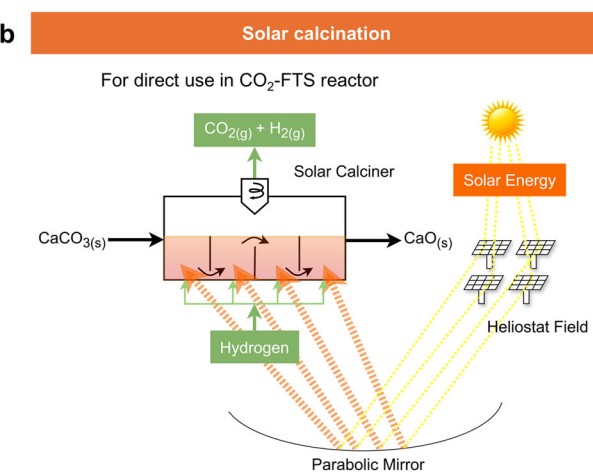

**Fig. 2 | Comparison of natural gas combustion-based calcination and the solar calcination. a** Natural gas combustion-based calcination (black) as used in Carbon Engineering's DAC process. Oxygen is separated using an air separation unit (ASU) for fluidisation, with heat supplied by natural gas combustion. **b** Solar calcination (orange) is proposed for the solar-driven DAC process. Solar calciner is fluidised by hydrogen and powered by concentrated solar energy. The mixed gas ($CO_2$ and $H_2$) from the solar calciner can be directly utilised in the $CO_2$-FTS reactor.

the front wall of the solar reactor using a heliostat field and parabolic mirror. This process cannot be directly integrated with DAC because the presence of fluidised air during the $CO_2$ desorption stage would render the $CO_2$ capture process ineffective.

Here, we show a large-scale route to produce SAF from air using solar-driven liquid-based direct air capture (L-DAC) and direct $CO_2$-FTS processes (Fig. 1b). The proposed L-DAC process uses a solar calciner with hydrogen as the fluidisation medium (Fig. 2b). This is different from the CE's design, which uses a natural gas combustion-based calciner with oxygen as the fluidisation medium (Fig. 2a)[22]. The DACCU process involves a tailored one-step $CO_2$-FTS process using Fe-Mn-K catalyst to produce jet fuel range hydrocarbons ($C_8$–$C_{16}$). The application of the $H_2$-fluidised solar calciner in the DAC process also acts synergistically to provide $H_2$ as feedstock for $CO_2$ hydrogenation. Compared to the previous stepwise DACCU process, the proposed process eliminates steps such as syngas production, $H_2$ preparation and $CO_2$ purification, making it easier to operate and more cost-effective. A comprehensive techno-economic assessment (TEA) has been conducted for the large-scale DACCU based on five possible locations worldwide. A roadmap for achieving future cost reduction is provided, demonstrating the potential for commercialisation of the technology to policymakers and industry investors.

## Results and discussion
### Solar-driven DACCU process and model assumptions
Figure 1b depicts a simplified process flow diagram (PFD) of the 1Mt$CO_2$ per year solar-driven DACCU process for SAF production. It consists of two main sections: (a) solar-driven DAC and (b) $CO_2$-to-SAF. The solar-driven section captures $CO_2$ from the air and consists of four major units, namely the air contactor, the pellet reactor, the slaker and the solar calciner. The $CO_2$-to-SAF section enables $CO_2$ utilisation to produce SAF through $CO_2$-FTS and the separation process. The whole process model was developed in Aspen Plus and used to explore how the assumptions and process requirements impact the process economics. Technical parameters for the base case, the optimistic and the pessimistic scenarios are given in Table 1.

The solar-driven DAC section is based on two closed chemical loops of K-cycle absorption (Eqs. (1) and (2)) and Ca-cycle desorption (Eqs. (3) and (4)). $CO_2$ in the air is driven by the fans into the air contactor (40,000 m² cross-section area) packed with Brentwood XF 12560 structured packing and captured by lean KOH solvent (2.0 M K⁺, 1.0 M OH⁻ and 0.5 M CO₃²⁻)[46–48]. A $CO_2$ capture rate of 74.5% was

achieved at an air travel distance (ATD) of 7 m and 1.4 m/s air velocity. Subsequently, the $CO_2$-rich $K_2CO_3$ solvent is crystallised with 30 wt% $Ca(OH)_2$ slurry in the bubbling fluidised pellet reactor. $CaCO_3$ pellet seeds are fed from the top of the bed, so the pellets are grown from the top until finished and discharged as large spherical pellets at the bottom[22]. After the $CaCO_3$ pellets are dried in the slaker and preheated in the two cyclones, they are decomposed to release the captured $CO_2$ and recover CaO at high-temperatures in the solar calciner.

$$2KOH_{(aq)} + CO_{2(g)} \rightarrow K_2CO_{3(l)} + H_2O_{(l)} \quad (1)$$

$$K_2CO_{3(aq)} + Ca(OH)_{2(s)} \rightarrow 2KOH_{(aq)} + CaCO_{3(s)} \quad (2)$$

$$CaCO_{3(s)} \rightarrow CaO_{(s)} + CO_{2(g)} \quad (3)$$

$$CaO_{(s)} + H_2O_{(g)} \rightarrow Ca(OH)_{2(s)} \quad (4)$$

The solar calciner is a four-stage horizontal hydrogen-fluidised bed reactor[45]. The $CaCO_3$ particles are fluidised by $H_2$ and decomposed using heat from CSP. This hydrogen-based fluidised bed approach has been successfully applied in the green steel industry for hydrogen direct reduction[49–51]. For instance, the MIDREX $H_2$™ project utilises 100% hydrogen as a reducing agent to manufacture iron, demonstrating the feasibility and effectiveness of hydrogen-based fluidised beds[52]. In our design, the heliostats field receives the direct normal irradiation (DNI) and reflects onto a parabolic mirror, which focuses the solar thermal energy onto the front wall of the solar calciner[45]. Heat is transferred from the front wall to the particles through radiation, conduction and convection, providing the sensible heat and enthalpy for the endothermic calcination reaction. The solar calciner is assumed to operate steadily at 813 °C with a $CaCO_3$ conversion of 95.2% based on pilot operating conditions[45]. The $H_2$ is assumed to be purchased from an off-site $H_2$ production plant with a stable supply.

Due to the intermittent nature of solar energy and the fact that DNI is available for only a fraction of the day—largely dependent on the sun's position and weather conditions such as clouds and fog—the solar calcination process and $CO_2$-to-SAF section are designed to operate as batch processes. These batch processes are scaled up to a much larger capacity than typically required for nominal production, resulting in a larger solar field and reactor size with respect to their

**Table 1 | Technical parameters for the solar-driven DAC and $CO_2$ utilisation (DACCU) plant for the base case, optimistic and pessimistic scenarios**

| Technical and design parameters | Units | Optimistic scenario | Base case scenario | Pessimistic scenario | Source |
|---|---|---|---|---|---|
| $CO_2$ capture capacity | Mt/yr | 0.96 | 0.96 | 0.96 | Process model |
| SAF productivity | kt/yr | 141.8 | 123.4 | 52.1 | Process model |
| Plant lifetime | year | 40 | 30 | 20 | 19 |
| Yearly operating hours for continuous process (L-DAC) | hr | 8000 | 8000 | 8000 | 19 |
| Yearly operating hours for the intermittent processes (solar calcination and $CO_2$-to-SAF) | hr | 3200 | 2667 | 2286 | Process model |
| Weighted average cost of capital | % | 5 | 10 | 15 | 18 |
| $CO_2$ capture rate | % | ~90% | ~75% | ~50% | Process model |
| $CO_2$ concentration in the air | ppm | 450 | 420 | 400 | 22 |
| Air velocity | m/s | 2 | 1.4 | 1 | 22 |
| Air travel distance | M | 11.7 | 7 | 3.5 | 22 |
| Dimensions of single air contactor (length × width × depth) | M | 5 × 5 × 11.7 | 5 × 5 × 7 | 5 × 5 × 3.5 | 22 |
| Solar multiple | N/A | 2.5 | 3 | 3.5 | Process model |
| Thermal efficiency of solar calciner | % | 80% | 60% | 40% | 63 |
| Dimensions of single solar calciner (length×width×bed height) | M | 13.4 × 1.1 × 5.3 | 13.4 × 1.1 × 5.3 | 13.4 × 1.1 × 5.3 | Process model |
| Maximum size of a single solar calciner | $MW_{th}$ | 40 | 40 | 40 | 45 |
| Number of solar calciners | N/A | 15 | 18 | 21 | Process mode |
| CAPEX of CSP | US$M | 189.5 | 379.0 | 568.4 | Process model |
| Gas recovery ratio | % | 99 | 90 | 80 | Process model |
| Total $CO_2$ conversion | mol % | 98.2 | 85.8 | 75.4 | Process model |
| Total jet fuel yield | mol % | 44.2 | 38.6 | 33.9 | Process model |
| SAF market price | US$/kg | 1.24 | 2.47 | 3.71 | 59 |
| Catalyst cost | US$/g | 3.98 | 3.98 | 3.98 | Estimated based on the cost of elements |
| Total land use | $km^2$ | 6.44 | 7.64 | 8.84 | Process model |
| Land cost | US$/$m^2$ | 1.24 | 2.47 | 49.42 | Process model |
| $H_2$ production cost | US$/kg | 1 | 2 | 3 | 61 |
| $H_2$ transportation cost | US$/kg | 0.18 | 0.18 | 0.18 | 62 |
| Pipeline required to transport $H_2$ | km | 50 | 50 | 50 | 62 |
| Electricity demand for DAC | MW | 11.1-20.2 | 11.9 | 8.2-12.5 | Process model |
| Electricity demand for $CO_2$ use | MW | 75.7-91.3 | 90.9 | 90.5-106.0 | Process model |
| PV electricity price | US$/MWh | 10 | 30 | 60 | 18 |

nominal capacity. This approach allows the solar calcination process to utilise sunlight for 7–10 h per day, depending on location. During this time, it can regenerate the CaO solids for the 24-h operation of the L-DAC process (see Supplementary Fig. 1). Additionally, the $CO_2$-to-SAF section can instantly convert gas products (i.e., $CO_2$ and $H_2$) from the solar calcination process into liquid products during sunlight hours. To facilitate flexible operation throughout the day, solid storage tanks are used for the storage of the high-temperature $CaCO_3$ and CaO particles. These storage tanks are designed with the capacity to support a full day of production[53,54]. Such high-temperature particle storage technology has been developed and shows less than 1% thermal loss per day[55]. This innovation addresses the intermittency of solar energy, eliminating the need for thermal energy storage systems and gas storage facilities. Under base case design conditions (a solar multiple of 3, which corresponds to 8 h of sunlight per day), eighteen 40 $MW_{th}$ (th refers to thermal energy) solar calciners, along with 2831 $m^3$ $CaCO_3$ storage and 1225 $m^3$ of CaO storage, would be necessary to maintain material balance with upstream and downstream processes.

In the $CO_2$-to-SAF section, the $CO_2$ and $H_2$ produced from the solar calciner are mixed with additional $H_2$ to achieve an $H_2$:$CO_2$ ratio of 3:1[28]. The direct $CO_2$-FTS process consists of the RWGS reaction

(Eq. (5)) and FTS reactions (Eqs. (6–8)) in a single reactor to produce SAF ($C_8$–$C_{16}$) and by-products such as gaseous hydrocarbons ($C_1$–$C_4$), liquid hydrocarbons ($C_5$–$C_7$) and wax ($C_{17+}$). The $CO_2$-FTS reactor is operated at 300 °C and 10 bar and catalysed by Me-Fe-K to achieve 38.2% $CO_2$ conversion and 47.8% selectivity to $C_8$–$C_{16}$ hydrocarbons[28]. The syncrude obtained from the $CO_2$-FTS reactor requires upgrading through separations and distillations to yield commercial products. Given that distillation systems are well-established in petroleum refining, similar equipment designs and operating conditions in prior studies can be adopted[56,57]. The produced SAF can be made ready for use by adding appropriate fuel additives or blending it with conventional jet fuel[58]. In this preliminary design, detailed modelling of the co-product separation system, the hydrocracking of heavy hydrocarbons and the blending process are not considered, while process improvement is employed through ex-situ water removal coupled with recirculation of unreacted $CO_2$, CO and $H_2$ to the $CO_2$-FTS reactor[35].

$$CO_2 + H_2 \rightleftharpoons CO + H_2O \quad (5)$$

$$nCO + (2n+1)H_2 \rightarrow C_nH_{2n+2} + nH_2O \quad (6)$$

$$nCO + 2nH_2 \rightarrow C_nH_{2n} + nH_2O \qquad (7)$$

$$nCO + 2nH_2 \rightarrow C_nH_{2n+1}OH + (n-1)H_2O \qquad (8)$$

## Baseline TEA

In the base case scenario, the 1MtCO$_2$/yr solar-driven DACCU plant can capture approximately 0.96 Mt CO$_2$ in the air and produce ~0.12 Mt SAF, which equals 50% of global SAF production in 2022[59]. Such large-scale solar-driven DACCU plants will be crucial for the aviation industry to meet its net-zero commitments by 2050[60]. The minimum selling price (MSP) of SAF is estimated at US\$4.62/kg, which is 1.9 times the 2022 market price (US\$2.4/kg) of SAF and 4.2 times that of conventional jet fuel (US\$1.1/kg)[59]. Detailed MSP cost breakdown, illustrating the capital and operational contributions, is shown in Fig. 3a. The levelized cost of solar-driven DACCU (LCOD) is projected at US\$283/t$_{CO2}$ (Fig. 3b), indicating the investment required to capture and convert each ton of atmospheric CO$_2$ to SAF, serving as a key indicator for policymakers providing incentives towards market success. However, it remains significantly above the industry target of US\$100/t$_{CO2}$[3].

It is evident from the results of final cost metrics that operational expenditure (OPEX) is the primary economic contributor, while capital expenditure (CAPEX) accounts for one-third of the overall costs. As illustrated in Fig. 3d, the CAPEX breakdown indicates that US\$1355 M (65.2%) of it is allocated to the solar-driven DAC plant and US\$703 M (33.8%) of it is allocated to the CO$_2$-to-SAF plant. The major equipment costs include the air contactor and the pellet reactor for DAC, the solar calciner and heliostat field for solar calcination, and compressors and CO$_2$-FTS reactor for CO$_2$-to-SAF. Detailed CAPEX information is summarised in Supplementary Tables 17 and 18. The annual OPEX for the plant is estimated to be US\$350 M (Fig. 3c), as detailed in Supplementary Tables 19 and 20. Notably, the OPEX is largely driven by the cost of hydrogen, which includes a production cost of US\$2.0/kg[61] and a transportation cost of US\$0.18/kg[62].

## Sensitivity analysis under optimistic and pessimistic scenarios

To gain a better understanding of potential cost reductions in the solar-driven DACCU process, we performed a single-variable sensitivity analysis on key variables in each section of the process, as well as for the entire process (see Table 1 for optimistic and pessimistic scenarios). In doing so, we will enhance the in-depth understanding of process operations and highlight the most important factors to overcome to enable commercial success.

The CO$_2$ capture productivity of the DAC plant is influenced by several operating and design variables, including the CO$_2$ concentration in the air, air velocity (V$_{air}$) and air travel distance (ATD). Figure 4a maps the CO$_2$ capture productivity at varying CO$_2$ concentrations (400 ppm to 450 ppm), V$_{air}$ (1 to 2 m/s) and CO$_2$ capture rates of ~50%, ~75% and ~90% with ATD controlled at 3.5, 7 and 11.7 meters. Under these conditions, the commercial-scale DAC plant can capture between 55.9 and 216.9 tonnes of CO$_2$ per hour, which has significant implications for energy and material consumption, ultimately influencing the final costs. The CO$_2$ capture rate is also influenced by climate-related conditions, including temperature and relative humidity (RH) (Supplementary Fig. 20), as reported in previous studies[18]. In this study, the base case assumes ambient conditions of 21 °C and 64% RH. A sensitivity analysis was conducted over a temperature range of 0 °C–30 °C and an RH range of 20%–80%. The results indicate that temperature has a greater impact than RH, with warm and humid conditions being the most favourable when considering cost implications.

The total hydrogen flow rate varies with solar calciner size, which in turn impacts the fluidisation conditions (Fig. 4b). As this is still in the early design stage, the cost of the solar calcination process remains inherently uncertain, relying on economic evaluations from CSP plants. The sensitivity analysis examines three main factors: the thermal efficiency of the solar calciner ($\eta_{th}$), the solar multiple (SM) and the capital cost of the CSP plant. The base case assumes a 60% thermal efficiency[39,63], with sensitivity scenarios at 40% and 80%. The baseline SM is set at 3, with variations tested at 2.5 and 3.5. To account for uncertainty in the CAPEX of the CSP, we vary the CSP CAPEX by ±50%. These analyses provide critical insights into the cost dynamics and optimisation potential of the CSP-DAC system.

The CO$_2$-to-SAF process applied an ex-situ water removal approach associated with gas recycling to improve CO$_2$ conversion and SAF yield. Figure 4c projects the improvements in CO$_2$ conversion and SAF yield at different gas recovery ratios. Without gas recovery, potential SAF and co-products from unreacted H$_2$, CO$_2$ and CO are wasted, resulting in an MSP of US\$11.64/kg and an LCOD of US\$502/tCO$_2$. Maximising gas recovery significantly improves product revenue, underscoring its importance if technology permits. Furthermore, the heat surplus in this CO$_2$-to-SAF process (Supplementary Fig. 21) can be utilised to offset the heating demand, leading to a 2.5% reduction in the MSP to US\$4.51/kg.

Based on the process variables investigated, the summary of economic sensitivity analysis results of the MSP and LCOD are illustrated in Fig. 4d, e. It was found that the H$_2$ production cost and weighted average cost of capital (WACC) are the primary cost drivers. Reducing the hydrogen production cost to US\$1/kg results in MSP decreasing to US\$3.50/kg (Fig. 4d) and LCOD dropping to US\$138/t$_{CO2}$ (Fig. 4e). Notably, the market price of SAF is the dominant factor for LCOD due to its cost-compensation effect. Other key factors include the gas recycle ratio, land cost, CAPEX of CSP, PV electricity price and thermal efficiency of solar calciner, which show considerable variability in their impact on the MSP and LCOD. Parameters such as plant lifetime, air velocity, gas recycle ratio, and solar multiple exhibit smaller impacts but remain integral to the overall cost structure.

## Geographical analysis

The TEA further investigates the impact of geographic locations on key factors such as land occupation and hydrogen production costs. This analysis selects five locations across different continents based on their high DNI and suitability for large-scale CSP plants. As shown in Fig. 5a, the selected countries capable of supporting large-scale CSP plants are limited to latitudes below 45°[64,65]. These regions, which are also suitable for solar PV, include the USA (North America), Chile (South America), Spain (Europe), South Africa (Africa) and China (Asia). The software System Advisor Model was employed to estimate land use requirements based on regional solar irradiation and daily sunlight hours. Chile, the USA and China demonstrate lower land use requirements, needing 6.94, 7.64 and 8.51 km$^2$, respectively. The extensive uninhabited areas in these regions make them suitable for the deployment of solar-driven DACCU plants.

The cost of purchased hydrogen in this proposed DACCU plant emerges as a key factor, as revealed by the sensitivity analysis. This cost exhibits significant variability across different countries and hydrogen production technologies. To minimise environmental impact, this analysis focuses on low-carbon hydrogen derived from several advanced technologies: alkaline electrolyser (AE), proton exchange membrane (PEM), solid oxide electrolysis cell (SOEC) and steam methane reforming (SMR) with carbon capture and storage (CCS). The study shows marked regional differences in the LCOD and MSP, which are heavily influenced by local hydrogen production costs and WACC (Fig. 5b–e). Under local WACC conditions (4.2%–11.8%)[66], China demonstrates the lowest MSP when using hydrogen from SMR with CCS (US\$3.23–3.79/kg$_{SAF}$). In contrast, Spain presents a cost advantage for AE (US\$4.15–5.84/kg$_{SAF}$), PEM (US\$4.82–6.51/kg$_{SAF}$) and SOEC (US\$6.17–8.42/kg$_{SAF}$) technologies. When evaluating the

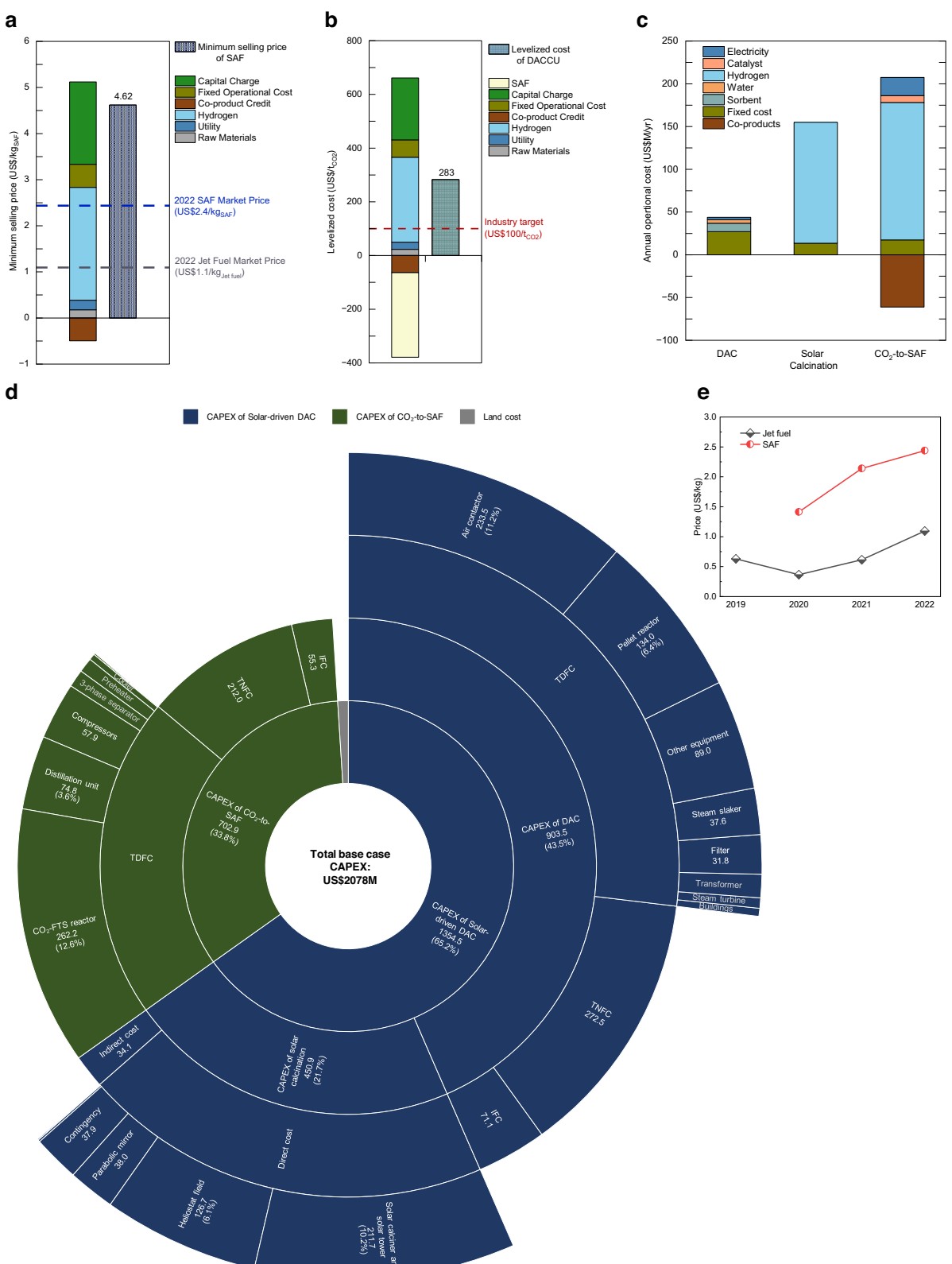

**Fig. 3 | Detailed cost breakdown of solar-driven DAC and CO₂ utilisation (DACCU) in the base case. a** Minimum selling price (MSP) of sustainable aviation fuel (SAF). **b** Levelized cost of the proposed solar-driven DACCU process. **c** Annual operational cost, disaggregated by process step. **d** Total capital expenditure (CAPEX), divided into two main sections: solar-driven DAC and CO₂-to-SAF. The DAC and solar calcination are subsections of the solar-driven DAC. **e** Global average market price of SAF and jet fuel. TDFC total direct field costs, IFC indirect field costs, TNFC total non-field costs.

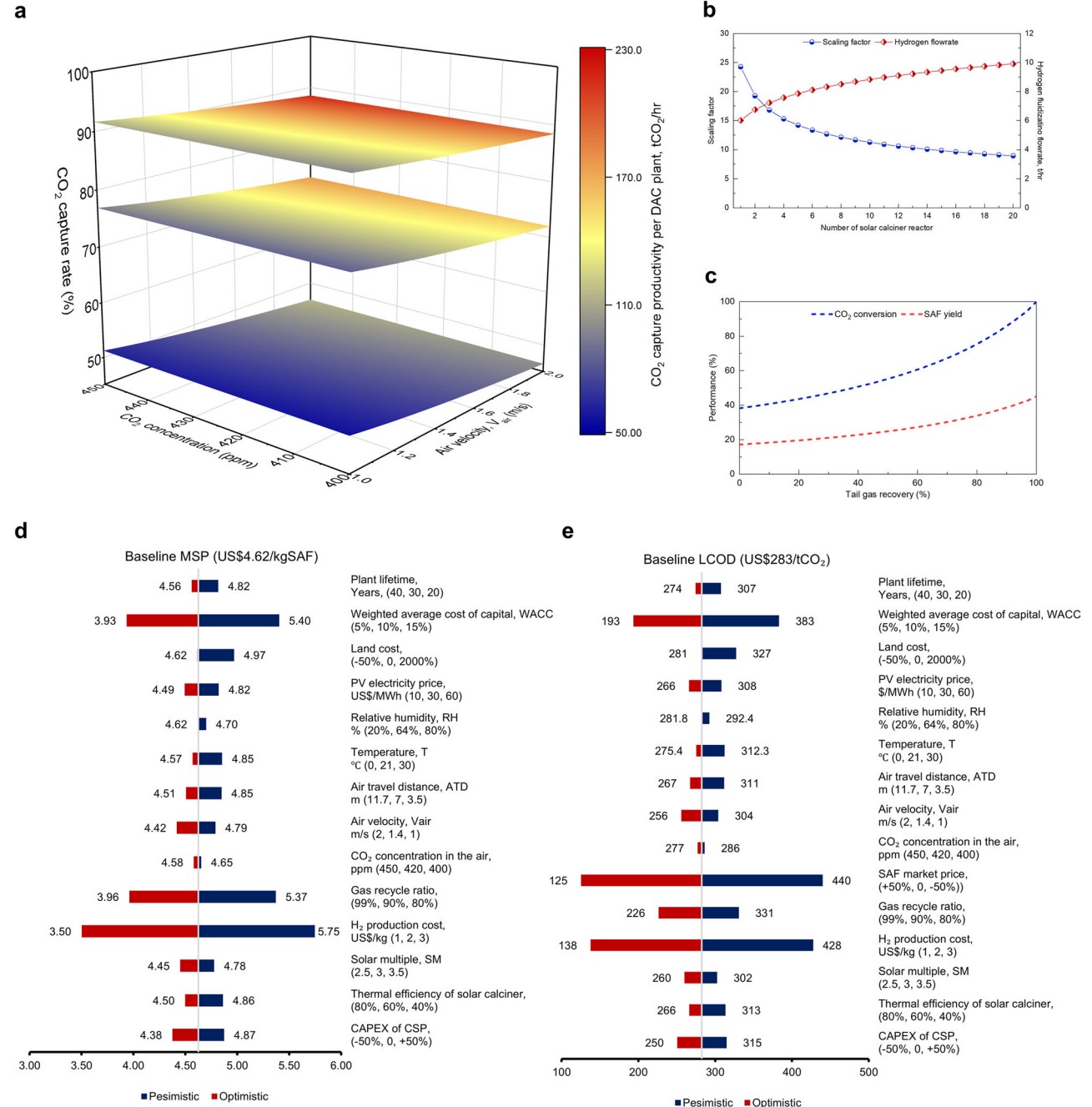

**Fig. 4 | Sensitivity analysis results for process parameters. a** Map of DAC plant $CO_2$ capture productivity. The $CO_2$ productivity as a function of $CO_2$ concentration, $V_{air}$ and ATD. Three coloured layers represent $CO_2$ capture rate at around 50% (ATD = 3.5 m), 75% (ATD = 7 m) and 90% (ATD = 11.7 m). **b** The impact of the scaling factor on the number of solar calciners and total hydrogen flow rate. **c** Process improvement on $CO_2$ conversion and sustainable aviation fuel (SAF) yield under

gas recovery rate from 0 to 99%. **d, e** Single variable sensitivity analysis of baseline cost for (**d**) minimum selling price (MSP) of SAF and (**e**) levelized cost of DAC and $CO_2$ utilisation (DACCU). The pessimistic and optimistic scenarios are depicted by red and blue bars, respectively, with the baseline cost represented by the central line.

plant cost under a global average WACC of 4.2%, previous low local WACC countries such as the USA and Spain lose their competitive edge. For hydrogen produced via SMR with CCS, the lowest MSP is attained in China (US\$2.92/kg$_{SAF}$) while South Africa (US\$3.14/kg$_{SAF}$) surpasses both the USA (US\$3.45/kg$_{SAF}$) and Spain (US\$3.25/kg$_{SAF}$). These findings underscore the substantial potential for cost reductions in solar-driven DACCU through the strategic selection of optimal deployment locations, particularly in regions with high solar irradiance, warm and humid climate, low land costs and favourable financial conditions.

## Comparison with previous studies

**Comparison with Carbon Engineering's DAC.** The proposed solar-driven DAC (CSP-DAC) process demonstrates improvements in terms of electricity demand and overall efficiency compared to CE's natural gas combustion-based DAC (NG-DAC). The CSP-DAC process shows a 63.0% reduction in electricity demand (267 kWh/t$_{CO_2}$) compared to NG-DAC (Fig. 6a). This reduction is primarily due to the elimination of the air separation unit (ASU) and lower $CO_2$ compression pressures. In the NG-DAC process, $CO_2$ is compressed to 151 bar for transport and storage, whereas in the continuous utilisation scenario, $CO_2$ is

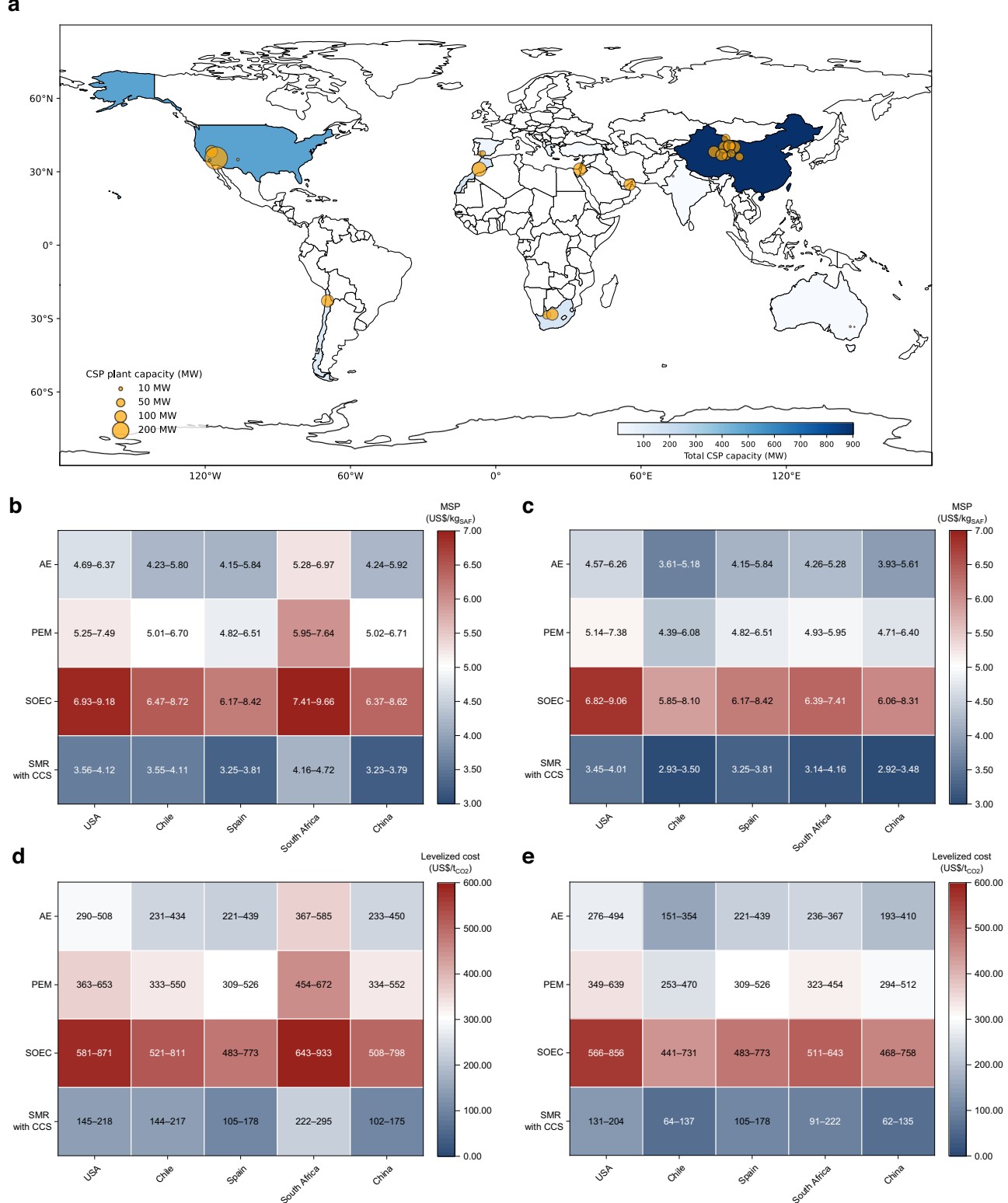

**Fig. 5 | Geographical analysis results for solar-driven direct air capture and CO₂ utilisation (DACCU).** **a** Global map of current high-temperature concentrated solar power (CSP) plants. Each bubble represents an individual plant, with the bubble size proportional to its installed capacity (MW), based on data from the SolarPACES[64]. **b, c** Minimum selling price (MSP) of sustainable aviation fuel (SAF) and **d, e** levelized cost of DACCU, with low-carbon hydrogen sourced at country-specific prices. **b** and **d** use local weighted average cost of capital (WACC); **c** and **e** use global averaged WACC of 4.2%. AE alkaline electrolyser, PEM proton exchange membrane, SOEC solid oxide electrolysis cell, SMR with CCS steam methane reforming with carbon capture and storage.

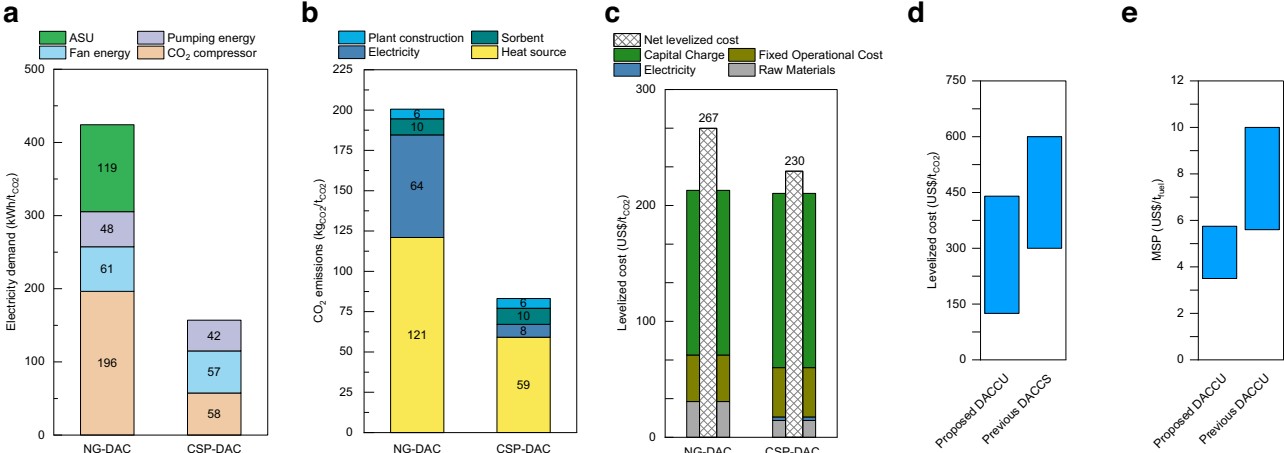

Fig. 6 | **Comparison analysis with previous DAC. a–c** Comparison between natural gas combustion-based DAC (NG-DAC) and the proposed solar-driven DAC (CSP-DAC): **a** electricity demand, **b** life cycle $CO_2$ emissions, and **c** levelized cost. **d**, **e** Comparison of proposed DAC and $CO_2$ utilisation (DACCU) with previous DAC and $CO_2$ storage (DACCS) and DACCU studies in terms of **d** levelized cost and **e** minimum selling price (MSP).

compressed only to 10 bar, which is also lower than the typical pressure for syngas production (30 bar).

Despite reduced electricity demand, DAC remains energy-intensive, with calcination being the primary energy consumer. The novel CSP-DAC plant can be self-sustaining, as the CSP provides the required heat, eliminating the need for onsite natural gas combustion. For CSP-DAC, the thermal energy requirement is 7.16 GJ/$t_{CO_2}$, assuming a solar calciner thermal efficiency of 60%. This is higher than the 5.52 GJ/$t_{CO_2}$ required by CE's DAC, which operates with a natural gas combustion-based calciner at 89% thermal efficiency.

From the preliminary life cycle assessment (LCA), CSP-DAC produces 58.5% fewer $CO_2$ emissions (117 kg$_{CO_2}$/$t_{CO_2}$) compared to NG-DAC (Fig. 6b). This reduction aligns with previous LCA studies and is primarily due to the shift to low-carbon energy sources[25]. The reduction in life cycle $CO_2$ emissions is mainly attributed to a decrease of 62 kg$_{CO_2}$/$t_{CO_2}$ from heat sources and an additional 56 kg$_{CO_2}$/$t_{CO_2}$ from the use of solar electricity.

In terms of cost, although the CAPEX for the CSP-DAC plant (US$1355 M) is higher than that of CE's DAC plant (~US$1200 M)[18,22], the net levelized cost of CSP-DAC (US$230/$t_{CO_2}$) is lower than NG-DAC (US$267/$t_{CO_2}$) (Fig. 6c). This cost advantage is due to the higher net carbon removal efficiency of CSP-DAC (91.7%) compared to NG-DAC (79.9%). As a result, the proposed CSP-DAC is not only cost-effective but also suited for the direct utilisation of air-captured $CO_2$.

**Comparison with DACCS**. When $CO_2$ captured from the air is intended for storage, the additional cost of transportation and storage increases the total expenses. A recent assessment by IEAGHG estimates the DACCS projects, which consider $CO_2$ capture, transport, and storage, will likely have levelized costs ranging from approximately US$300 to 600 per ton of $CO_2$ stored, based on global average solar PV costs[18]. In contrast, the proposed solar-driven DACCU pathway achieves a lower levelized cost range (US$138–428/$t_{CO_2}$) as shown in Fig. 6d, while also avoiding the technological and economic uncertainties associated with $CO_2$ transport and storage. The cost advantage is primarily due to the combination of $CO_2$ utilisation to produce value-added SAF, which helps offset total costs. Moreover, there is potential for profitability if the revenue generated from the $CO_2$ utilisation process exceeds the overall costs.

**Comparison with stepwise DACCU**. Previous synthetic fuel production through DAC and FTS pathways typically includes three stages: DAC, syngas production, and FTS. In contrast, the proposed process

bypasses the syngas production stage entirely and eliminates the need for $CO_2$ purification and $H_2$ preparation since the mixed gas ($CO_2$ and $H_2$) produced from the solar calciner can be directly used for downstream processes. This streamlining makes the proposed process more cost-effective compared to previous stepwise DACCU processes. For example, Rojas-Michaga et al. reported the MSP of jet fuel at US$6.55/kg$_{jet}$ for a solid-based DAC with $CO_2$ utilisation[12]. Similarly, Marchese et al. accessed a CE-based DAC with $CO_2$ utilisation for wax production, with MSP ranging from US$5.6 to 10.0/kg$_{wax}$ depending on plant configurations[12]. These costs are substantially higher than our proposed process (Fig. 6e), where the MSP is only US$4.62/kg$_{SAF}$ at the base case and ranges from US$3.50 to 5.75/kg$_{SAF}$ under optimistic and pessimistic scenarios.

**A roadmap predicting cost reduction potential**

In this paper, the base case represents the first-of-a-kind (FOAK) plants and is assumed to be deployed in the near term. However, the estimated costs are high with existing technology and market conditions. Here, we present a detailed roadmap (Fig. 7) for achieving a more competitive cost reduction for $N^{th}$-of-a-kind (NOAK) plants through a waterfall analysis, illustrating the cumulative repercussions of various process advancements. The MSP of SAF for the NOAK plant could be reduced to US$2.12/kg, which is below the current market price of US$2.4/kg. The LCOD could decrease to −US$47/$t_{CO_2}$, indicating that the entire capture and utilisation process is profitable. As revealed from the single-variable sensitivity and geographical analyses, the cost-effective hydrogen production technology is prioritised as the initial step in the roadmap. Implementing these changes could eliminate more than 24% of the total cost for MSP of SAF and 51% for LCOD.

Subsequent technological advancements are essential to improve the efficiency of DAC, CSP, and $CO_2$-to-SAF processes, thereby offsetting the total cost. Key factors include enhancing the gas recycle ratio in $CO_2$-FTS, increasing the thermal efficiency of the solar calciner in CSP, and optimising $CO_2$ capture efficiency in DAC. Besides, further studies on waste gas recycling and wax upgrading can boost total co-product credits[67]. Additionally, reducing the PV electricity price for the entire process shows potential for further cost reductions. Considering that hydrogen is produced off-site and purchased, its cost is not directly impacted by on-site electricity prices. However, the low price of renewable electricity significantly affects both the DACCU process and hydrogen production. Thus, securing low-cost renewable electricity is critical for overall economic viability.

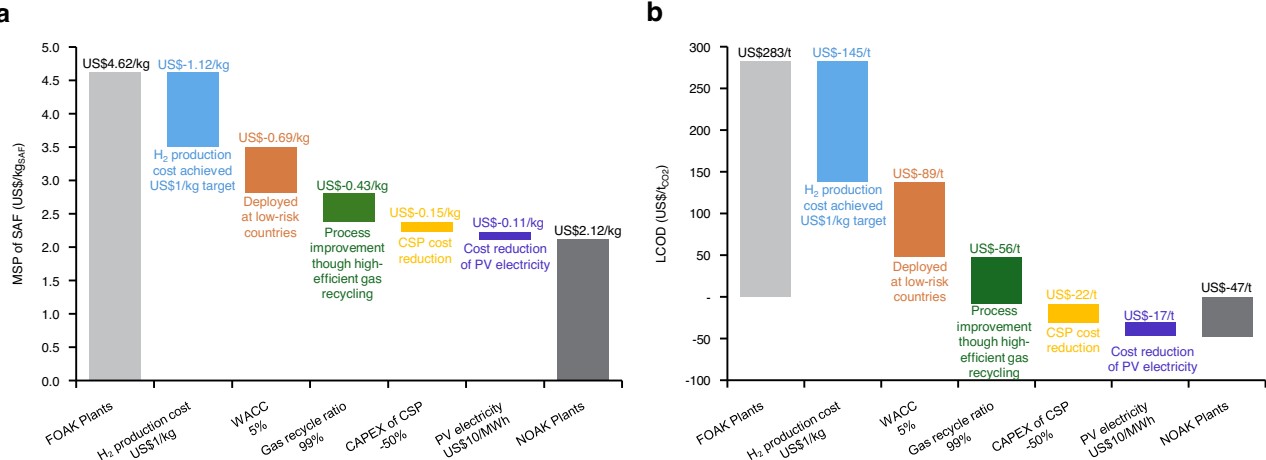

**Fig. 7 | Roadmap to reducing base case cost by successive changes to cost-relevant parameters from first-of-a-kind (FOAK) plants to N$^{th}$-of-a-kind (NOAK) plants.** **a** Minimum selling price (MSP) of sustainable aviation fuel (SAF). **b** levelized cost of DAC and $CO_2$ utilisation (DACCU).

Further cost reductions can be explored through government policies and incentives, such as carbon credits. The high carbon price can offset the costs and foster a robust carbon market, encouraging investment in DAC-based technologies. For instance, a higher carbon price above the levelized cost could make DAC or DACCU profitable. Programs like the 45Q project, which provides credits of US$180 per ton of $CO_2$ permanently stored and US$130 per ton for $CO_2$ used[68], could significantly impact the economics of DAC projects. Lastly, promoting industry-academia collaborations and public-private partnerships will drive innovation and facilitate the sharing of best practices.

## Limitations and perspectives

This study presents a comprehensive design for an environmentally attractive and cost-effective large-scale solar-driven DACCU process aimed at producing SAF. The proposed process is designed to operate at a scale of 1 MtCO$_2$/yr and has been developed through modelling, simulation, validation and scale-up.

The TEA and preliminary LCA demonstrate the advantages of the solar-driven DAC process. Key benefits include a 63% reduction in electricity consumption (267 kWh/t$_{CO2}$) and a 59% reduction in process $CO_2$ emissions (117 kg$_{CO2}$/t$_{CO2}$) compared to the DAC process by CE.

Additionally, the proposed process is also cost-effective compared to previous processes because (i) the levelized cost is US$283/t$_{CO2}$, which is cheaper than the ~US$300-600/t$_{CO2}$ range reported for DACCS; (ii) the MSP of US$4.62/kg is lower than the MSP obtained through the stepwise DACCU process, which ranges from US$5.6 to 10.0/kg.

A sensitivity analysis indicates that the hydrogen production cost and WACC are the two major cost drivers affecting the MSP of SAF. Furthermore, a geographical analysis highlights the regional impact on the global feasibility of such a solar-driven plant.

We also predicted the potential for cost reduction through a roadmap for future plants. The SAF produced from solar-driven DACCU plants could become more cost-competitive and even profitable in the future if: (a) plants are built in locations with cost-effective hydrogen production technology and low WACC, (b) significant technological advancements are made across all sections, and (c) supportive policies, such as carbon credits are introduced.

Due to the limited data availability and the early-stage development of solar-driven DACCU technology, assumptions were made for the scale-up of the $CO_2$ utilisation process and the process design of the $H_2$ production plant and downstream upgrading. Further real-world, large-scale, one-step $CO_2$-FTS plant assessments are essential to

ensure practical feasibility. Future experimental studies are necessary to validate the produced aviation fuel's physicochemical properties and ensure compliance with full certification requirements for neat or high-blend use. Moreover, the impact of variable climate conditions on renewable hydrogen production and electrolyser design should be considered to better assess the stability and operability of the entire process. In addition, detailed process design of the distillation system for co-product recovery could further enhance the overall plant economics.

Moreover, the proposed process opens avenues for further research into developing alternative fluidisation mediums, efficient solar calciner designs and novel $CO_2$-to-SAF catalysts. Implementing effective heat recovery and water integration strategies would help reduce energy consumption and operational costs[69]. Comprehensive cradle-to-grave LCA studies are necessary to quantify environmental impacts and ensure compliance with stringent environmental standards. The worldwide or nationwide potential could be further explored, as tailored regional operating strategies may exist. Additionally, socio-political analysis is vital for understanding the implications of deploying large-scale DACCU plants, which will facilitate broader adoption through social acceptance, regulatory support, and policy incentives.

## Methods
### Process model development and validation

A process model for solar-driven DACCU was developed in Aspen Plus® V11 to explore the FOAK plant productivity and economic performance under achievable design conditions. This model is justified by the validation of different key streams and units and subsequent scaling-up. Figure 1b and Supplementary Fig. 1 depict the PFD for the base case and the detailed process flow information, indicating all model inputs are provided in Supplementary Figs. 2–4.

### Simulation and comparison of solar-driven DAC process at commercial scale

Process simulation of DAC is initially carried out, and the results are compared with CE's open commercial simulation and subsequently adapted to a solar-driven DAC configuration. We use the RK-SOAVE, ENTRTL-RK and SOLIDS thermodynamic property packages for the gaseous phase, aqueous phase and solid phase, respectively.

In terms of air contactor modelling[70], we incorporate a modified built-in packing to represent the Brentwood XF12560 packing, and the packing pressure drop is adjusted based on correlations derived from pilot experiments. The simulation results, as outlined in the

Supplementary Note 2, demonstrate significant agreement on material and energy balance. This strong agreement provides us with confidence in replacing the natural gas combustion-based calciner with a solar calciner.

By incorporating the solar calciner, we can eliminate the need for additional units such as an ASU, a $CO_2$ absorber unit, and a water knockout. Despite the retrofitting of the calciner, the proposed solar-driven DAC process maintains the same capacity of 1 $MtCO_2$/yr. This is due to the retention of the air contactor unit with its original air inlet area. The simulation of the solar-driven DAC process is conducted based on the design wherein the solar calciner model replaces the calciner model in the simulated DAC process.

**Modelling, simulation and validation of solar calciner at pilot scale.**
The pilot scale solar reactor was modelled as a 1D steady-state four-stage horizontal fluidised bed, which was implemented in Aspen Plus® V11 using SOLIDS physical property linked with Aspen Custom Modeller® (ACM) to correct the stream enthalpy and process thermal efficiency. Two representative experimental datasets are used for validation. Simulation results align with expectations, as detailed in Supplementary Note 3.

**Modelling, simulation and validation of $CO_2$-FTS at lab scale.** The $CO_2$-to-FTS process was simulated in Aspen Plus® V11 using the Peng-Robinson physical property method. We employed the modified Anderson-Schulz-Flory theory to predict the hydrocarbon distributions for SAF production through a direct $CO_2$-FTS approach. The hydrocarbon distributions were validated based on the experimental selectivity of CO, $C_1$, $C_2$–$C_4$, $C_{5+}$, and $C_8$–$C_{16}$. The relative errors between model prediction and experimental data of product selectivity in targeted carbon ranges were below 0.8%. Then, hydrocarbon distributions were represented by lumping components and $CO_2$-FTS reactions were listed by representative reactions. Since the selectivity towards oxygenated compounds is below 1.0% during experiments[28], they were neglected in this model. Hence, only olefins and paraffins were considered. The detailed modelling, simulation and validation procedures can be found in the Supplementary Note 4.

**Scale-up method**
The scaling law in open literature was adopted for the scaling of solar calciner, which is considered the most efficient and cost-effective method for determining the hydrodynamics of a hot fluidised bed system[71,72]. It should be noted that some studies also considered the scaling effect on the reaction[73]. Here, the scale-up approach of solar calciner considers both hydrodynamics and chemical conversion[74]. The detailed scale-up approach is described in the Supplementary Note 5. The scaling factor is based on the commercial size of the solar calciner. At a specific size, the design and operating parameters when using hydrogen as a fluidisation medium are determined with the scaling law.

The $CO_2$-FTS reactor and Fe-Mn-K catalyst are assumed to behave the same way at lab-scale and large scale[35]. Therefore, the operating conditions are the same at different scales, and the impact of the reactor dimensions on the reactions is neglected. The material and energy flow of the large-scale $CO_2$ utilisation process is simulated based on validated and scaled models.

**TEA**
In this study, the high-level TEA is carried out to highlight the cost drivers and geographical impacts toward the successful deployment of the proposed process. Supplementary Fig. 14 summarises the key input and output parameters of the model. Parameters such as temperature, RH, DNI, SM and WACC are more regionally dependent, whereas parameters such as gas recovery ratio, scaling factors, and reactor efficiency are technology-dependent in the model. In practice,

some of these factors would show regional variation as well, for instance, the cost of PV electricity price according to the risk premium of countries, but this global TEA does not consider these regional influences.

**CAPEX.** Based on the material flow and energy requirement, the equipment size and cost are determined, from which the total CAPEX is estimated based on the literature reported method[17,22,75].

The CAPEX of the DAC plant and $CO_2$-to-SAF plant are calculated based on Eqs. (9)–(11).

$$Total\ field\ cost = Field\ cost + Non-field\ cost \qquad (9)$$

$$Direct\ field\ cost = \sum Installed\ equipment\ cost \qquad (10)$$

$$Installed\ equipment\ cost = Equipment\ cost \times Installation\ factor \qquad (11)$$

However, the literature studies do not emphasise the economics of CSP-based solar calcination. Considering that this technology is at the preliminary design stage, we estimated the cost of CSP based on literature-reported CSP technologies such as the parabolic trough, concentrated solar power tower and beam-down solar concentrating[40,63,76–78]. The evaluated equipment of CSP includes the heliostat field, parabolic mirror, solar calciner and tower. It should be noted that the CAPEX of CSP was considered for the deliberately scaled solar calcination process, and the size of the storage tanks was calculated based on the flowrate of solids[41,54].

The CAPEX of the CSP plant is calculated based on Eqs. (12)–(14)[39].

$$CAPEX = Direct\ cost + Indirect\ cost \qquad (12)$$

$$Direct\ cost = Contingency + Factor \times Direct\ cost \qquad (13)$$

$$Indirect\ cost = Land\ cost + Factor \times Indirect\ cost \qquad (14)$$

**OPEX.** The TEA assumes 8000 operating hours for the continuous process per year for economic evaluation[18]. The fixed OPEX includes maintenance, labour, administration, and other costs. The annual fixed OPEX is assumed to be 3% of the CAPEX. The variable OPEX covers electricity consumption, co-product credits and material inputs (i.e., sorbent, water, catalyst and hydrogen). The input information obtained from the model on raw material (e.g., KOH and $CaCO_3$), hydrogen, electricity, etc., was used to estimate the annual variable OPEX. For TEA analysis under the base, optimistic and pessimistic scenarios, the green hydrogen is produced off-site in an alkaline electrolyser (AE) plant located 50 km away from the DAC plant and transported through a 10-inch diameter pipeline. Considering the hydrogen is purchased for use, the cost of hydrogen, including production and transportation, is US$2.18/$kg_{H2}$[61,62]. The electricity demand for fans, pumps, compressors and heaters is assumed to be supplied by the PV system to minimise environmental impact. Makeup materials such as KOH, $CaCO_3$, and water are added based on the mass balance of the process model. The sorbent price is assumed at US$750/$t_{KOH}$ and US$/200$t_{CaCO3}$. The industrial water price is assumed at US$1/$m^3$ as the average price for the five studied locations[12,17,18]. Given the early-stage design, a detailed estimation of the co-product recovery system's OPEX was not included, as it is expected to be relatively low (<0.5%) compared to annual OPEX.

**Levelized cost and MSP.** The prediction of the CAPEX and OPEX enables the calculation of two cost metrics: (a) the levelized cost and

(b) the MSP. The equation for the levelized cost is provided as Eq. (15)[18].

$$Levelized\ cost = \frac{(CAPEX \times CRF + annual\ variable\ OPEX + annual\ fixed\ OPEX)}{annual\ CO_2\ captrue}$$

(15)

With capital recovery factor (CRF) represents the portion of the initial CAPEX that needs to be paid every year. CRF is based on the weighted average cost of capital (WACC) and plant lifetime as shown in Eq. (16)[18].

$$CRF = \frac{WACC \times (1 + WACC)^{Lifetime}}{(1 + WACC)^{Lifetime} - 1}$$

(16)

This levelized cost, calculated this way, represents the cost of capturing and processing one tonne of $CO_2$ from the atmosphere. However, the construction or operation procedures emit $CO_2$ or other GHGs. The net levelized cost can be estimated based on carbon removal efficiency as defined in Eq. (17)[18].

$$Net\ levelised\ cost = \frac{Levelized\ cost}{carbon\ removal\ efficiency}$$

(17)

The carbon removal efficiency[79] defined in Eq. (18), is the percentage of net $CO_2$ captured from air in the lifecycle.

$$Carbon\ removal\ efficiency = 1 - \frac{total\ LCA\ emissions}{total\ CO_2\ captured\ from\ air}$$

(18)

The equation for the MSP of SAF is provided as Eq. (19).

$$MSP = \frac{(CAPEX \times CRF + annual\ variable\ OPEX + annual\ fixed\ OPEX)}{annual\ SAF\ production}$$

(19)

**Sensitivity analysis.** To understand the impact of key parameters on overall solar-driven DACCU cost, we carried out a sensitivity analysis on TEA. The impact of different operating and design variables on each sector (i.e., solar-driven DAC, solar calcination and $CO_2$-to-SAF) and financial accounting parameters was investigated. This cost is not optimised from every variable connected with the final economic analysis, which is far beyond the preliminary design stage.

**Preliminary LCA**

In this paper, the environmental benefits of using solar energy to power DAC are examined by a preliminary LCA. We calculate the plant construction emissions, sorbent production emissions, and energy-related (heat and electricity) emissions[18]. Note that this study does not perform a full cradle-to-grave LCA analysis and relies on publicly available sources for estimating emissions. The LCA analysis is only carried out on the solar-driven DAC process to have a clear view of the $CO_2$ emissions cut when using renewable energy to replace natural gas. Any potential $CO_2$ emissions from the $CO_2$ utilisation plant are not included.

## Data availability

The data supporting the findings of this study are available within the article or Supplementary Information file. Source data are provided with this paper.

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

## Acknowledgements

This work was supported by EU RISE project OPTIMAL (Ref: 101007963 to Y.H. and M.W.), National Key Research & Development Program—Intergovernmental International Science and Technology Innovation Cooperation Project (Ref: 2021YFE0112800 to W.D. and F.Q.) and National Natural Science Foundation of China—Basic Science Centre Programme (Ref: 61988101 to W.D. and F.Q.).

## Author contributions

Conceptualisation: Y.H., O.O., M.W. and F.Q.; Methodology: Y.H., O.O., A.D.N.K., M.W. and W.D.; Software: Y.H., O.O., A.D.N.K., H.Y. and M.W.; Writing–original draft: Y.H.; Writing–review & editing: Y.H., O.O., A.D.N.K., H.Y., F.M., M.W., W.D. and F.Q.; Supervision: M.W., W.D. and F.Q.

## Competing interests

The authors declare no competing interests.
