## [Transparent Peer Review file · Nature Communications]

Solar-driven direct air capture to produce sustainable aviation fuel

Corresponding Author: Professor Meihong Wang

Version 0:

Reviewer comments:

Reviewer #1

(Remarks to the Author)

I believe my previous comments have been adequately addressed. This is a well-executed study, and I support its publication.

Reviewer #2

(Remarks to the Author)

This study proposes a solar-driven DACCU process as a potential pathway for SAF production. While the paper presents a rigorous simulation and techno-economic analysis of the entire process, it appears to be a system-level integration of several works, some of which were already published by the same group. As such, the study lacks sufficient originality for publication in Nature Communications. Additionally, the following key issues require clarification.

1. Table 1 - Electricity Demand for DAC: The pessimistic scenario reports a lower electricity demand than the optimistic scenario, which is counterintuitive. The authors should justify this finding by explaining the underlying mechanisms driving electricity demand variations across scenarios.
2. Additional evidence is needed to validate whether the produced SAF meets commercial aviation fuel standards and demand. This is critical for assessing the practical applicability of the proposed technology.
3. If no distillation equipment is designed, it is inappropriate to calculate the prices of by-products solely based on the prices of pure substances. The article does not provide a detailed description of the distillation system used for separating by-products. From the given operating and equipment costs, the system may not be able to fully separate the co-products. In particular, separating propylene from propane and ethylene from ethane is highly energy-intensive, and this should be considered.
4. While concentrated solar power and storage tanks are proposed to mitigate intermittency, the study does not sufficiently address how renewable hydrogen production—highly sensitive to climate conditions—is stabilized for continuous operation. Further elaboration is needed.
5. The assumed hydrogen price appears overly optimistic, significantly influencing the overall process economics. A more conservative estimate, aligned with current renewable hydrogen market trends, should be provided.

Reviewer #3

(Remarks to the Author)

The authors have responded to the comments provided in the review of the previous version of the manuscript. I find that the responses are adequate. However, I have a few minor comments:

1. In Figure 2a, the input to the calciner should be CaCO_3
2. In line MW(th), what does th stand for?
3. In Fig. 5, kindly explain in more detail how plant scale is interpreted.

4. Please discuss the limitations of your work in the concluding section of the manuscript.

Version 1:

Reviewer comments:

Reviewer #3

(Remarks to the Author)

My comments have been addressed and I have nothing further to add.

Reviewer #4

(Remarks to the Author)

I was asked to judge if the authors had satisfactorily addressed Reviewer #2's prior technical concerns.

In comment 1, to address the concerns about "counterintuitive" results on electricity demands under different scenarios, the authors have provided additional discussion on the underlying mechanism and revealed clearly the impact of air velocity, air travel distance, and CO₂ concentrations. These explanations make the results no longer "counterintuitive".

In comment 2, the authors have now provided adequate examples, including documents showing industrial standards and real-world applications by commercialized companies, to prove that the SAF produced through FT synthesis meets commercial aviation fuel standards and demand, clearly demonstrating the practical applicability of their technology.

In comment 3, the authors have now provided a detailed analysis of the distillation process and the cost for by-product treatment. The additionally provided data sufficiently revealed that the total installed cost of the distillation unit accounts for 5.4% of the total DACCU plant cost, and the annual cost for electricity and solvent only accounts for a minor part of the total OPEX, for example, 0.26% for electricity. I think these additional discussions, illustrating figures, and quotations adequately solve Reviewer #2's concerns about distillation.

In comment 4, the authors have now clarified that the hydrogen is purchased directly and that the supply is considered steady. They also provided techno-economic implications of hydrogen price variability through sensitivity and regional analyses. The impact of climate conditions on hydrogen production is, as stated by the author, outside the scope of this paper.

In comment 5, the authors have provided adequate examples and evidence to prove the rationality of the hydrogen price.

Given the above analysis, I believe the authors have fully addressed Reviewer #2's technical concerns.

A point-by-point response to the reviewers' comments

Black colour: the comments from reviewers.

Blue colour: response to the comments from reviewers.

Red colour: revised and new sentences in the revised manuscript.

Reviewer #1 (Remarks to the Author):

I believe my previous comments have been adequately addressed. This is a well-executed study, and I support its publication.

Response: We sincerely thank the reviewer for their positive feedback and continued support for the publication of our work.

Reviewer #2 (Remarks to the Author):

This study proposes a solar-driven DACCU process as a potential pathway for SAF production. While the paper presents a rigorous simulation and techno-economic analysis of the entire process, it appears to be a system-level integration of several works, some of which were already published by the same group. As such, the study lacks sufficient originality for publication in Nature Communications. Additionally, the following key issues require clarification.

Response: We sincerely thank the reviewer for spending time to evaluate our work and for providing insightful comments that will help us improve the quality of the manuscript. We are particularly excited to see the reviewer's observation regarding the "rigorous simulation and techno-economic analysis of the entire process". It's a system-level integration of several components, but the novelties lie in:

- (a) We proposed an **H₂-fluidised solar calciner** for the L-DAC process with continuous and intermittent operations.

This innovation enables a fully solar-powered L-DAC process, distinguishing it from previously reported works on L-DAC processes.

- (b) We designed a tailored **intermittent** one-step CO₂-FTS process that directly utilises the **mixed** gas stream of CO₂ and H₂ produced by the solar-driven L-DAC process.

This approach differs from our earlier published work, which focused on a **continuous** CO₂-FTS process that was fed by **separately sourced** CO₂ and H₂ streams.

- (c) System-level analysis of different **geographical locations** considering **climate conditions** and **technical performance**.

The feasibility of the proposed first-of-a-kind (FOAK) DACCU plants was evaluated in the USA, Chile, Spain, South Africa and China.

The remaining technical comments have been addressed in detail in the following responses.

1. Table 1 - Electricity Demand for DAC: The pessimistic scenario reports a lower electricity demand than the optimistic scenario, which is counterintuitive. The authors should justify this finding by explaining the underlying mechanisms driving electricity demand variations across scenarios.

Response: Thank you for your insightful comments. We understand the reviewer’s concerns regarding the lower electricity demand for the pessimistic scenario compared to the optimistic scenario. The underlying mechanisms are detailed in Table S28 of the Supplementary Information (see Table below).

Table S28: Sensitivity analysis of parameters for DAC—material requirement and electricity consumption under different operating conditions

Parameter	Pr	conc	V _{air}	ATD	CL	F _{air}	F _{KOH}	R _{KOH}	F _{Ca(OH)₂}	F _{water}	Fan ΔP	E _{fan}	E _{pump}	E _{DAC}	F _{CaCO₃}
Unit	t-CO ₂ /h	ppm	m/s	m	%	kt/h	kt/h	%	t/h	t	Pa	kWh/t-CO ₂	kWh/t-CO ₂	kWh/t-CO ₂	t/h
Base case	119.4	420	1.4	7	74.5	251.0	32.6	99.7	716.5	516.1	85.7	57.4	42.1	99.5	238.7
400 ppm	113.7	400	1.4	7	74.5	251.0	32.6	99.7	682.4	489.7	85.7	60.3	44.2	104.5	271.7
450 ppm	128.0	450	1.4	7	72.0	251.0	32.6	99.7	767.7	555.8	85.7	53.6	39.3	92.9	305.7
1m/s V _{air}	88.0	420	1	7	76.9	179.3	23.3	99.7	527.7	369.8	43.0	27.9	40.9	68.8	210.1
2m/s V _{air}	164.9	420	2	7	74.5	358.6	46.6	99.7	989.1	727.4	181.4	125.8	43.6	169.4	393.8
3.5m ATD	78.3	420	1.4	3.5	48.9	251.0	16.3	99.7	469.6	324.7	51.0	52.1	32.1	84.2	187.0
11.7m ATD	144.4	420	1.4	11.7	90.1	251.0	54.5	99.7	866.4	632.3	126.1	69.8	58.2	128.1	345.0

Electricity demand for DAC is influenced by a combination of key parameters, including air velocity (V_{air}), CO₂ concentration (conc), and air travel distance (ATD). These parameters would affect air flow rate (F_{air}), liquid flow rates (F_{KOH}, F_{Ca(OH)₂} and F_{water}) and CO₂ capture productivity (Pr), and consequently, fan and pump electricity demand.

In the three **optimistic** scenarios:

- A higher air velocity (V_{air} = 2 m/s) increases the volume of air processed and thus requires higher solvent flow rates and greater fan and pump power.
- A longer air travel distance (ATD = 11.7 m) requires more liquid to wet the packing column, thereby increasing pump energy.
- A higher CO₂ concentration (conc=450 ppm) *reduces* electricity demand due to improved capture productivity.

In the three **pessimistic** scenarios:

- A lower air velocity (V_{air} = 1 m/s) reduces the air throughput and solvent demand, thereby lowering fan and pump power.
- A shorter air travel distance (ATD = 3.5 m) decreases the liquid for wetting the packing column, leading to reduced pump energy.

- A lower CO₂ concentration (conc=400 ppm) *increases* electricity demand due to reduced capture productivity.

Action: We have added the explanation for the underlying mechanisms on Page 33 of the Supplementary Information. We have carefully re-examined the electricity consumption data and updated it in Table 1 of the revised manuscript.

2. Additional evidence is needed to validate whether the produced SAF meets commercial aviation fuel standards and demand. This is critical for assessing the practical applicability of the proposed technology.

Response: We appreciate the reviewer’s comment regarding the applicability of the produced SAF to meet commercial aviation fuel standards. The synthetic aviation fuel produced in our proposed system is consistent with Fischer–Tropsch Synthetic Paraffinic Kerosene (FT-SPK), which is a recognised drop-in SAF pathway approved under Advancing Standards Transforming Markets (ASTM D7566). According to current ASTM guidelines, FT-SPK can be blended with conventional jet fuel up to 50% for use in commercial aviation without requiring modifications to engines or infrastructure.

Moreover, industry demonstrations (1POINTFIVE fuel synthesis) have shown real-world applicability of such fuels. For example:

- 1PointFive’s FT-SPK has been approved for blending up to 50% with Jet A or Jet A-1, based on ASTM D7566 specifications.
- Carbon Engineering and Air To Fuels™ have also demonstrated co-processing routes where FT crude is blended at levels up to 5% directly with conventional jet fuel without further upgrading.

These examples confirm that the synthetic hydrocarbons produced via FT synthesis—like those in our process—are compatible with existing aviation systems when blended within approved limits. While our current work focuses on the system model of SAF production via solar-driven DACCU, the resulting product is suitable for blending with conventional jet fuel at this early stage of technology development. We agree with the reviewer that further experimental validation of fuel properties, particularly for neat or high-blend usage, will be essential in future work.

Action: We have added the new sentences to Lines 452-454 (Page 22) of the revised manuscript to reflect this limitation and future work.

3. If no distillation equipment is designed, it is inappropriate to calculate the prices of by-products solely based on the prices of pure substances. The article does not provide a detailed description of the distillation system used for separating by-products. From the given operating and equipment costs, the system may not be able to fully separate the co-products. In particular, separating propylene from propane and ethylene from ethane is highly energy-intensive, and this should be considered.

Response: Thank you for the thoughtful comments. We agree that evaluating the prices of co-products requires considering the cost of distillation equipment needed for product separation. In our study, we have indeed evaluated and costed a distillation system for co-product recovery. As illustrated in the Extended Data Fig.4 on page 26 of the revised manuscript, two distillation processes are presented: one for the separation of SAF and wax, and another for the recovery of co-products.

While the co-product recovery is illustrated as a single block in Extended Data Fig. 4, for clarity, this unit consists of multiple components, including distillation columns, compressors, heaters and coolers. Specifically, the process handles a stream of 55.8 t/h, which contains 10.1 wt% water, 69.3 wt% C₁-C₄ hydrocarbons and 20.4 wt% C₅⁺ hydrocarbons. After water removal, the remaining hydrocarbons can be separated by conventional petroleum refinery processes—a mature and well-established industrial process. Here, we adopt the same separation process from previous study (*Energy Environ. Sci.* 16, 3638–3653; [DOI: [10.1039/D3EE00749A](https://doi.org/10.1039/D3EE00749A)]) to recover high-purity (>99.5 wt%) fractions of CH₄, C₂H₄, C₂H₆, C₃H₆, C₃H₈, C₄H₈, C₄H₁₀ and C₅-C₇ naphtha (see Figure below).

Figure. Distillation process for co-product recovery

The cost of this separation system is included in Table S17 of the Supplementary Information. The total installed cost of the distillation unit is **\$74.8 million**, which represents approximately **5.4% of the total DACCU plant cost**. Of this, **\$44.3 million (~59.3%)** is allocated specifically to co-product recovery. This value is based on a bottom-up estimate of major equipment costs for separation, summarised in the table below.

Table. Major installed equipment cost breakdown of co-product recovery

Equipment	Equipment Cost (2020)	Installed Cost (2020)	Installation factor	Saling exponent	Ref Equipment cost	Ref Installed cost	Model Capacity	Ref Capacity	Unit	Base year	Ref
Demethanizer	1,456,373	2,907,122	2.00	0.65	259,000	517,000	50	4.06	t/h	2016	Yadav et al. (2021)
Deethanizer	1,176,874	2,010,493	1.71	0.65	240,000	410,000	38.7	3.88	t/h	2016	Yadav et al. (2021)
C2 splitter	1,506,794	2,495,627	1.66	0.65	256,000	424,000	9.40	0.71	t/h	2016	Yadav et al. (2021)
Depropanizer	756,281	1,414,525	1.87	0.65	162,000	303,000	29.3	3.17	t/h	2016	Yadav et al. (2021)
C3 Fractionator	4,253,699	5,526,370	1.30	0.65	1,113,000	1,446,000	12.1	1.78	t/h	2016	Yadav et al. (2021)
Debutanizer	423,637	1,112,754	2.63	0.65	75,000	197,000	17.2	1.39	t/h	2016	Yadav et al. (2021)
Extractive distillation	1,158,607	1,378,886	1.19	0.65	405,000	482,000	5.8	1.33	t/h	2016	Yadav et al. (2021)
Stripper	260,619	353,698	1.36	0.65	126,000	171,000	2.7	1.02	t/h	2016	Yadav et al. (2021)
Subtotal	\$17,199,475										
Total installed cost of co-product process (major equipment and other equipment) = \$17,199,475 * 2.58 = \$44,322,392											

Regarding operating costs, we recognise that separating close-boiling-point hydrocarbons such as ethylene/ethane and propylene/propane is energy intensive. Accordingly, we have considered the cost of electricity consumption, which is estimated based on the data from the referenced study (*Energy Environ. Sci.* 16, 3638–3653; [DOI: [10.1039/D3EE00749A](https://doi.org/10.1039/D3EE00749A)]). In our analysis, the **annual electricity cost** for the entire co-product recovery system is estimated to be **\$0.89 million (0.26% of the total OPEX of the DACCU process)**. The cost for solvent consumption, such as DMF used for extractive distillation of C₄ hydrocarbons can be neglected in this case, which is only \$0.04 million per year. Given this relatively minor contribution to operating expenses and the early-stage nature of the DACCU system, we have not included a detailed breakdown of OPEX for co-product recovery at this time. Our objective in presenting co-product revenue is to emphasize the potential of co-product recovery as a strategy for offsetting costs. Therefore, based on our assumption regarding the process and cost estimates for the co-product distillation systems, we believe it is appropriate to estimate by-product revenues using the market prices of the primary separated products. We also acknowledge that future work should pursue a more detailed techno-economic analysis of the separation process, including partial recoveries, purity constraints, and dynamic pricing considerations.

Action: We have clarified in Lines 211-213 (Page 8) of the revised manuscript that the standard industrial distillation process could be adopted. Additionally, we have included references to previous studies that offer detailed cost breakdowns and equipment specifications for co-product distillation systems. The limitations of this simplified design approach are now acknowledged on Line 215 (Page 8) and further discussed in Lines 448-457 (Page 22) of the revised manuscript.

We have clarified the assumption regarding the OPEX of the distillation system on Lines 577-579 (Page 32) of the revised manuscript. Additionally, we have explained that the CAPEX for the distillation system, which includes distillation columns, separators, dryers and coolers, were estimated using a holistic approach as mentioned on Page 23 of the Supplementary Information. The detailed cost estimation for co-product recovery is illustrated in Figure S10 (Page 24) of the Supplementary Information.

4. While concentrated solar power and storage tanks are proposed to mitigate intermittency, the study does not sufficiently address how renewable hydrogen production—highly sensitive to climate conditions—is stabilized for continuous operation. Further elaboration is needed.

Response: Thank you for raising this important point. In our study, the central focus is on the CSP-driven DAC and CO₂ utilisation process for SAF production. To ensure process continuity, we assumed a steady hydrogen supply from an off-site hydrogen production plant located about 50 km away.

Although the operational stability of the external hydrogen plant is considered to be outside the scope and system boundary of the present work. However, we analysed the techno-economic implications of hydrogen price variability through sensitivity and regional analyses.

To reflect this limitation more transparently, we have also added a discussion in the conclusion section, noting that the effects of variable climate conditions on the hydrogen production costs are not considered in the current work and should be addressed in future system-level analyses.

Action: We have added a new assumption for steady hydrogen supply from the external hydrogen plant in Lines 183-184 (Page 8) of the revised Manuscript. We also added new texts for further elaboration in Lines 448-450 and Lines 454-456 (Page 22) of the revised Manuscript.

5. The assumed hydrogen price appears overly optimistic, significantly influencing the overall process economics. A more conservative estimate, aligned with current renewable hydrogen market trends, should be provided.

Response: We respect the reviewer's comments regarding the assumed hydrogen price. However, we believe our baseline assumption of **\$2/kgH₂** is justified based on both government targets and recent studies. Specifically, this value is consistent with the 2025 cost target outlined by the U.S. Department of Energy for electrolysis-based hydrogen production (DOE, 2024). In addition, a recent peer-reviewed study by Zuo et al. (2023) (*Nat. Commun.* 14, 4680; [DOI: 10.1038/s41467-023-40319-5]) reported a green hydrogen production cost of \$2.09/kg, closely matching our base case.

To further evaluate the economic sensitivity, we conducted a comprehensive sensitivity analysis identifying hydrogen price as a key cost driver. Consequently, we extended our analysis to include region-specific hydrogen pricing, reflecting a realistic market spread. In this geographical analysis, renewable hydrogen production cost varied from **\$1.8 to 6.5/kg**, depending on technology and location.

Furthermore, the IEA Global Hydrogen Review 2024 supports this cost range, indicating a projected renewable hydrogen production cost of approximately \$2-6/kg (see Figure below). Therefore, we believe that the assumed hydrogen price in our analysis is both reasonable and consistent with current market expectations.

[Figure Redacted]

Action: No action is needed for this comment.

Reviewer #3 (Remarks to the Author):

The authors have responded to the comments provided in the review of the previous version of the manuscript. I find that the responses are adequate. However, I have a few minor comments:

Response: We thank the reviewer for carefully reading our work. We appreciate your further points raised to further improve the quality of the manuscript.

1. In Figure 2a, the input to the calciner should be CaCO_3

Response: We thank the reviewer for catching this. Fig. 2a has been corrected accordingly. Please refer to Page 6 of our revised manuscript.

2. In line $\text{MW}(\text{th})$, what does th stand for?

Response: Thank you for your comment. The “ MW_{th} ” refers to megawatts of thermal energy. We have now defined this clearly. Please refer to Line 201 (Page 8) of the revised manuscript.

3. In Fig. 5, kindly explain in more detail how plant scale is interpreted.

Response: Thank you for your comment. In Fig. 5a, the plant scale refers to the actual capacity of currently operating large-scale concentrated solar power (CSP) plants around the world. To obtain this information, we used data from the SolarPACES database, which lists detailed specifications for commercial CSP plants. Each bubble in the figure represents a specific CSP plant, with the bubble size proportional to its installed capacity.

Action: To enhance clarity, we have revised the scale bar in Fig. 5a and added a more detailed explanation in the figure caption. This will help readers better interpret the bubble sizes in relation to plant scale. Please refer to Page 16 of our revised manuscript.

4. Please discuss the limitations of your work in the concluding section of the manuscript.

Response: We appreciate this suggestion and have now included the limitations in the conclusion section.

Action: We have changed the title of the “Conclusion and perspectives” section to “Conclusion, limitations and perspectives”. We have further discussed the limitations of (a) scale-up of the CO_2 utilisation process; (b) process design of H_2 production plant and distillation system for

co-product recovery; and (c) experimental validation for synthetic aviation fuel's physicochemical properties. Please refer to Lines 448-457 (Page 22) of the revised manuscript.